# Multi-way Interacting Regression via Factorization Machines

**Mikhail Yurochkin**
Department of Statistics
University of Michigan
moonfolk@umich.edu

**XuanLong Nguyen**
Department of Statistics
University of Michigan
xuanlong@umich.edu

**Nikolaos Vasiloglou**
LogicBlox
nikolaos.vasiloglou@logicblox.com

## Abstract

We propose a Bayesian regression method that accounts for multi-way interactions of arbitrary orders among the predictor variables. Our model makes use of a factorization mechanism for representing the regression coefficients of interactions among the predictors, while the interaction selection is guided by a prior distribution on random hypergraphs, a construction which generalizes the Finite Feature Model. We present a posterior inference algorithm based on Gibbs sampling, and establish posterior consistency of our regression model. Our method is evaluated with extensive experiments on simulated data and demonstrated to be able to identify meaningful interactions in applications in genetics and retail demand forecasting.[1]

## 1 Introduction

A fundamental challenge in supervised learning, particularly in regression, is the need for learning functions which produce accurate prediction of the response, while retaining the explanatory power for the role of the predictor variables in the model. The standard linear regression method is favored for the latter requirement, but it fails the former when there are complex interactions among the predictor variables in determining the response. The challenge becomes even more pronounced in a high-dimensional setting – there are exponentially many potential interactions among the predictors, for which it is simply not computationally feasible to resort to standard variable selection techniques (cf. Fan & Lv (2010)).

There are numerous examples where accounting for the predictors' interactions is of interest, including problems of identifying epistasis (gene-gene) and gene-environment interactions in genetics (Cordell, 2009), modeling problems in political science (Brambor et al., 2006) and economics (Ai & Norton, 2003). In the business analytics of retail demand forecasting, a strong prediction model that also accurately accounts for the interactions of relevant predictors such as seasons, product types, geography, promotions, etc. plays a critical role in the decision making of marketing design.

A simple way to address the aforementioned issue in the regression problem is to simply restrict our attention to lower order interactions (i.e. 2- or 3-way) among predictor variables. This can be achieved, for instance, via a support vector machine (SVM) using polynomial kernels (Cristianini & Shawe-Taylor, 2000), which pre-determine the maximum order of predictor interactions. In practice, for computational reasons the degree of the polynomial kernel tends to be small. Factorization machines (Rendle, 2010) can be viewed as an extension of SVM to sparse settings where most

interactions are observed only infrequently, subject to a constraint that the interaction order (a.k.a. interaction depth) is given. Neither SVM nor FM can perform any selection of predictor interactions, but several authors have extended the SVM by combining it with $\ell_1$ penalty for the purpose of feature selection (Zhu et al., 2004) and gradient boosting for FM (Cheng et al., 2014) to select interacting features. It is also an option to perform linear regression on as many interactions as we can and combine it with regularization procedures for selection (e.g. LASSO (Tibshirani, 1996) or Elastic net (Zou & Hastie, 2005)). It is noted that such methods are still not computationally feasible for accounting for interactions that involve a large number of predictor variables.

In this work we propose a regression method capable of adaptive selection of multi-way interactions of arbitrary order (MiFM for short), while avoiding the combinatorial complexity growth encountered by the methods described above. MiFM extends the basic factorization mechanism for representing the regression coefficients of interactions among the predictors, while the interaction selection is guided by a prior distribution on random hypergraphs. The prior, which does not insist on the upper bound on the order of interactions among the predictor variables, is motivated from but also generalizes Finite Feature Model, a parametric form of the well-known Indian Buffet process (IBP) (Ghahramani & Griffiths, 2005). We introduce a notion of the hypergraph of interactions and show how a parametric distribution over binary matrices can be utilized to express interactions of unbounded order. In addition, our generalized construction allows us to exert extra control on the tail behavior of the interaction order. IBP was initially used for infinite latent feature modeling and later utilized in the modeling of a variety of domains (see a review paper by Griffiths & Ghahramani (2011)).

In developing MiFM, our contributions are the following: (i) we introduce a Bayesian multi-linear regression model, which aims to account for the multi-way interactions among predictor variables; part of our model construction includes a prior specification on the hypergraph of interactions — in particular we show how our prior can be used to model the incidence matrix of interactions in several ways; (ii) we propose a procedure to estimate coefficients of arbitrary interactions structure; (iii) we establish posterior consistency of the resulting MiFM model, i.e., the property that the posterior distribution on the true regression function represented by the MiFM model contracts toward the truth under some conditions, without requiring an upper bound on the order of the predictor interactions; and (iv) we present a comprehensive simulation study of our model and analyze its performance for retail demand forecasting and case-control genetics datasets with epistasis. The unique strength of the MiFM method is the ability to recover meaningful interactions among the predictors while maintaining a competitive prediction quality compared to existing methods that target prediction only.

The paper proceeds as follows. Section 2 introduces the problem of modeling interactions in regression, and gives a brief background on the Factorization Machines. Sections 3 and 4 carry out the contributions outlined above. Section 5 presents results of the experiments. We conclude with a discussion in Section 6.

## 2    Background and related work

Our starting point is a model which regresses a response variable $y \in \mathbb{R}$ to observed covariates (predictor variables) $x \in \mathbb{R}^D$ by a non-linear functional relationship. In particular, we consider a multi-linear structure to account for the interactions among the covariates in the model:

$$\mathbb{E}(Y|x) = w_0 + \sum_{i=1}^{D} w_i x_i + \sum_{j=1}^{J} \beta_j \prod_{i \in Z_j} x_i. \tag{1}$$

Here, $w_i$ for $i = 0, \ldots, D$ are bias and linear weights as in the standard linear regression model, $J$ is the number of multi-way interactions where $Z_j, \beta_j$ for $j = 1, \ldots, J$ represent the interactions, i.e., sets of indices of interacting covariates and the corresponding interaction weights, respectively. Fitting such a model is very challenging even if dimension $D$ is of magnitude of a dozen, since there are $2^D - 1$ possible interactions to choose from in addition to other parameters. The goal of our work is to perform interaction selection and estimate corresponding weights. Before doing so, let us first discuss a model that puts a priori assumptions on the number and the structure of interactions.

## 2.1 Factorization Machines

Factorization Machines (FM) (Rendle, 2010) is a special case of the general interactions model defined in Eq. (1). Let $J = \sum_{l=2}^{d} \binom{D}{l}$ and $Z := \bigcup_{j=1}^{J} Z_j = \bigcup_{l=2}^{d} \{(i_1, \ldots, i_l)|i_1 < \ldots < i_l; i_1, \ldots, i_l \in \{1, \ldots, D\}\}$. I.e., restricting the set of interactions to $2, \ldots, d$-way, so (1) becomes:

$$\mathbb{E}(Y|x) = w_0 + \sum_{i=1}^{D} w_i x_i + \sum_{l=2}^{d} \sum_{i_1=1}^{D} \cdots \sum_{i_l=i_{l-1}+1}^{D} \beta_{i_1,\ldots,i_l} \prod_{t=1}^{l} x_{i_t}, \tag{2}$$

where coefficients $\beta_j := \beta_{i_1,\ldots,i_l}$ quantify the interactions. In order to reduce model complexity and handle sparse data more effectively, Rendle (2010) suggested to factorize interaction weights using PARAFAC (Harshman, 1970): $\beta_{i_1,\ldots,i_l} := \sum_{f=1}^{k_l} \prod_{t=1}^{l} v_{i_t,f}^{(l)}$, where $V^{(l)} \in \mathbb{R}^{D \times k_l}$, $k_l \in \mathbb{N}$ and $k_l \ll D$ for $l = 2, \ldots, d$. Advantages of the FM over SVM are discussed in details by Rendle (2010). FMs turn out to be successful in the recommendation systems setups, since they utilize various context information (Rendle et al., 2011; Nguyen et al., 2014). Parameter estimation is typically achieved via stochastic gradient descent technique, or in the case of Bayesian FM (Freudenthaler et al., 2011) via MCMC. In practice only $d = 2$ or $d = 3$ are typically used, since the number of interactions and hence the computational complexity grow exponentially. We are interested in methods that can adapt to fewer interactions but of arbitrarily varying orders.

# 3 MiFM: Multi-way Factorization Machine

We start by defining a mathematical object that can encode sets of interacting variables $Z_1, \ldots, Z_J$ of Eq. (1) and selecting an appropriate prior to model it.

## 3.1 Modeling hypergraph of interactions

Multi-way interactions are naturally represented by hypergraphs, which are defined as follows.

**Definition 1.** Given $D$ vertices indexed by $S = \{1, \ldots, D\}$, let $Z = \{Z_1, \ldots, Z_J\}$ be the set of $J$ subsets of $S$. Then we say that $G = (S, Z)$ is a hypergraph with $D$ vertices and $J$ hyperedges.

A hypergraph can be equivalently represented as an incidence binary matrix. Therefore, with a bit abuse of notation, we recast $Z$ as the matrix of interactions, i.e., $Z \in \{0, 1\}^{D \times J}$, where $Z_{i_1 j} = Z_{i_2 j} = 1$ iff $i_1$ and $i_2$ are part of a hyperedge indexed by column/interaction $j$.

Placing a prior on multi-way interactions is the same as specifying the prior distribution on the space of binary matrices. We will at first adopt the Finite Feature Model (FFM) prior (Ghahramani & Griffiths, 2005), which is based on the Beta-Bernoulli construction: $\pi_j|\gamma_1, \gamma_2 \overset{iid}{\sim} \text{Beta}(\gamma_1, \gamma_2)$ and $Z_{ij}|\pi_j \overset{iid}{\sim} \text{Bernoulli}(\pi_j)$. This simple prior has the attractive feature of treating the variables involved in each interaction (hyperedge) in an symmetric fashion and admits exchangebilility among the variables inside interactions. In Section 4 we will present an extension of FFM which allows to incorporate extra information about the distribution of the interaction degrees and explain the choice of the parametric construction.

## 3.2 Modeling regression with multi-way interactions

Now that we know how to model unknown interactions of arbitrary order, we combine it with the Bayesian FM to arrive at a complete specification of MiFM, the Multi-way interacting Factorization Machine. Starting with the specification for hyperparameters:

$$\sigma \sim \Gamma(\alpha_1/2, \beta_1/2), \qquad \lambda \sim \Gamma(\alpha_0/2, \beta_0/2), \qquad \mu \sim \mathcal{N}(\mu_0, 1/\gamma_0),$$
$$\lambda_k \sim \Gamma(\alpha_0/2, \beta_0/2), \qquad \mu_k \sim \mathcal{N}(\mu_0, 1/\gamma_0) \text{ for } k = 1, \ldots, K.$$

Interactions and their weights:

$$w_i|\mu, \lambda \sim \mathcal{N}(\mu, 1/\lambda) \text{ for } i = 0, \ldots, D, \qquad Z \sim \text{FFM}(\gamma_1, \gamma_2),$$
$$v_{ik}|\mu_k, \lambda_k \sim \mathcal{N}(\mu_k, 1/\lambda_k) \text{ for } i = 1, \ldots, D; k = 1, \ldots, K.$$

Likelihood specification given data pairs $(y_n, x_n = (x_{n1}, \ldots, x_{nD}))_{n=1}^N$:

$$y_n | \Theta \sim \mathcal{N}(y(x_n, \Theta), \sigma), \text{ where } y(x, \Theta) := w_0 + \sum_{i=1}^D w_i x_i + \sum_{j=1}^J \sum_{k=1}^K \prod_{i \in Z_j} x_i v_{ik}, \quad (3)$$

for $n = 1, \ldots, N$, and $\Theta = \{Z, V, \sigma, w_{0,\ldots,D}\}$. Note that while the specification above utilizes Gaussian distributions, the main innovation of MiFM is the idea to utilize incidence matrix of the hypergraph of interactions $Z$ with a low rank matrix $V$ to model the mean response as in Eq. 1. Therefore, within the MiFM framework, different distributional choices can be made according to the problem at hand — e.g. Poisson likelihood and Gamma priors for count data or logistic regression for classification. Additionally, if selection of linear terms is desired, $\sum_{i=1}^D w_i x_i$ can be removed from the model since FFM can select linear interactions besides higher order ones.

## 3.3 MiFM for Categorical Variables

In numerous real world scenarios such as retail demand forecasting, recommender systems, genotype structures, most predictor variables may be categorical (e.g. color, season). Categorical variables with multiple attributes are often handled by so-called "one-hot encoding", via vectors of binary variables (e.g., IS_blue; IS_red), which must be mutually exclusive. The FFM cannot immediately be applied to such structures since it assigns positive probability to interactions between attributes of the same category. To this end, we model interactions between categories in $Z$, while with $V$ we model coefficients of interactions between attributes. For example, for an interaction between "product type" and "season" in $Z$, $V$ will have individual coefficients for "jacket-summer" and "jacket-winter" leading to a more refined predictive model of jackets sales (see examples in Section 5.2).

We proceed to describe MiFM for the case of categorical variables as follows. Let $U$ be the number of categories and $d_u$ be the set of attributes for the category $u$, for $u = 1, \ldots, U$. Then $D = \sum_{u=1}^U \text{card}(d_u)$ is the number of binary variables in the one-hot encoding and $\bigsqcup_{u=1}^U d_u = \{1, \ldots, D\}$. In this representation the input data of predictors is $X$, a $N \times U$ matrix, where $x_{nu}$ is an active attribute of category $u$ of observation $n$. Coefficients matrix $V \in \mathbb{R}^{D \times K}$ and interactions $Z \in \{0,1\}^{U \times J}$. All priors and hyperpriors are as before, while the mean response (3) is replaced by:

$$y(x, \Theta) := w_0 + \sum_{u=1}^U w_{x_u} + \sum_{k=1}^K \sum_{j=1}^J \prod_{u \in Z_j} v_{x_u k}. \quad (4)$$

Note that this model specification is easy to combine with continuous variables, allowing MiFM to handle data with different variable types.

## 3.4 Posterior Consistency of the MiFM

In this section we shall establish posterior consistency of MiFM model, namely: the posterior distribution $\Pi$ of the conditional distribution $P(Y|X)$, given the training $N$-data pairs, contracts in a weak sense toward the truth as sample size $N$ increases.

Suppose that the data pairs $(x_n, y_n)_{n=1}^N \in \mathbb{R}^D \times \mathbb{R}$ are i.i.d. samples from the joint distribution $P^*(X, Y)$, according to which the marginal distribution for $X$ and the conditional distribution of $Y$ given $X$ admit density functions $f^*(x)$ and $f^*(y|x)$, respectively, with respect to Lebesgue measure. In particular, $f^*(y|x)$ is defined by

$$Y = y_n | X = x_n, \Theta^* \sim \mathcal{N}(y(x_n, \Theta^*), \sigma), \text{ where } \Theta^* = \{\beta_1^*, \ldots, \beta_J^*, Z_1^*, \ldots, Z_J^*\},$$

$$y(x, \Theta^*) := \sum_{j=1}^J \beta_j^* \prod_{i \in Z_j^*} x_i, \text{ and } x_n \in \mathbb{R}^D, y_n \in \mathbb{R}, \beta_j^* \in \mathbb{R}, Z_j^* \subset \{1, \ldots, D\} \quad (5)$$

for $n = 1, \ldots, N, j = 1, \ldots, J$. In the above $\Theta^*$ represents the *true* parameter for the conditional density $f^*(y|x)$ that generates data sample $y_n$ given $x_n$, for $n = 1, \ldots, N$. A key step in establishing posterior consistency for the MiFM (here we omit linear terms since, as mentioned earlier, they can be absorbed into the interaction structure) is to show that our PARAFAC type structure can approximate arbitrarily well the true coefficients $\beta_1^*, \ldots, \beta_J^*$ for the model given by (1).

**Lemma 1.** Given natural number $J \geq 1$, $\beta_j \in \mathbb{R} \setminus \{0\}$ and $Z_j \subset \{1, \ldots, D\}$ for $j = 1, \ldots J$, exists $K_0 < J$ such that for all $K \geq K_0$ system of polynomial equations $\beta_j = \sum_{k=1}^K \prod_{i \in Z_j} v_{ik}, j = 1, \ldots, m$ has at least one solution in terms of $v_{11}, \ldots, v_{DK}$.

The upper bound $K_0 = J - 1$ is only required when *all* interactions are of the depth $D - 1$. This is typically not expected to be the case in practice, therefore smaller values of $K$ are often sufficient.

By conditioning on the training data pairs $(x_n, y_n)$ to account for the likelihood induced by the PARAFAC representation, the statistician obtains the posterior distribution on the parameters of interest, namely, $\Theta := (Z, V)$, which in turn induces the posterior distribution on the conditional density, to be denoted by $f(y|x)$, according to the MiFM model (3) without linear terms. The main result of this section is to show that under some conditions this posterior distribution $\Pi$ will place most of its mass on the true conditional density $f^*(y|x)$ as $N \to \infty$. To state the theorem precisely, we need to adopt a suitable notion of weak topology on the space of conditional densities, namely the set of $f(y|x)$, which is induced by the weak topology on the space of joint densities on $X, Y$, that is the set of $f(x, y) = f^*(x)f(y|x)$, where $f^*(x)$ is the true (but unknown) marginal density on $X$ (see Ghosal et al. (1999), Sec. 2 for a formal definition).

**Theorem 1.** Given any true conditional density $f^*(y|x)$ given by (5), and assuming that the support of $f^*(x)$ is bounded, there is a constant $K_0 < J$ such that by setting $K \geq K_0$, the following statement holds: for any weak neighborhood $U$ of $f^*(y|x)$, under the MiFM model, the posterior probability $\Pi(U|(X_n, Y_n)_{n=1}^N) \to 1$ with $P^*$-probability one, as $N \to \infty$.

The proof's sketch for this theorem is given in the Supplement.

## 4 Prior constructions for interactions: FFM revisited and extended

The adoption of the FFM prior on the hypergraph of interactions carries a distinct behavior in contrast to the typical Latent Feature modeling setting. In a standard Latent Feature modeling setting (Griffiths & Ghahramani, 2011), each row of $Z$ describes one of the data points in terms of its feature representation; controlling row sums is desired to induce sparsity of the features. By contrast, for us a column of $Z$ is identified with an interaction; its sum represents the interaction depth, which we want to control a priori.

**Interaction selection using MCMC sampler**     One interesting issue of practical consequence arises in the aggregation of the MCMC samples (details of the sampler are in the Supplement). When aggregating MCMC samples in the context of *latent feature modeling* one would always obtain exactly $J$ latent features. However, in *interaction modeling*, different samples might have no interactions in common (i.e. no exactly matching columns), meaning that support of the resulting posterior estimate can have up to $\min\{2^D - 1, IJ\}$ unique interactions, where $I$ is the number of MCMC samples. In practice, we can obtain marginal distributions of all interactions across MCMC samples and use those marginals for selection. One approach is to pick $J$ interactions with highest marginals and another is to consider interactions with marginal above some threshold (e.g. 0.5). We will resort to the second approach in our experiments in Section 5 as it seems to be in more agreement with the concept of "selection". Lastly, we note that while a data instance may a priori possess unbounded number of features, the number of possible interactions in the data is bounded by $2^D - 1$, therefore taking $J \to \infty$ might not be appropriate. In any case, we do not want to encourage the number of interactions to be too high for regression modeling, which would lead to overfitting. The above considerations led us to opt for a parametric prior such as the FFM for interactions structure $Z$, as opposed to going fully nonparametric. $J$ can then be chosen using model selection procedures (e.g. cross validation), or simply taken as the model input parameter.

**Generalized construction and induced distribution of interactions depths**     We now proceed to introduce a richer family of prior distributions on hypergraphs of which the FFM is one instance. Our construction is motivated by the induced distribution on the column sums and the conditional probability updates that arise in the original FFM. Recall that under the FFM prior, interactions are a priori independent. Fix an interaction $j$, for the remainder of this section let $Z_i$ denote the indicator of whether variable $i$ is present in interaction $j$ or not (subscript $j$ is dropped from $Z_{ij}$ to simplify notation). Let $M_i = Z_1 + \ldots + Z_i$ denote the number of variables among the first $i$ present in the corresponding interaction. By the Beta-Bernoulli conjugacy, one obtains $\mathbb{P}(Z_i = 1|Z_1, \ldots, Z_{i-1}) = \frac{M_{i-1} + \gamma_1}{i - 1 + \gamma_1 + \gamma_2}$. This highlights the "rich-gets-richer" effect of the FFM prior, which encourages the existence of very deep interactions while most other interactions have very small depths. In some situations we may prefer a relatively larger number of interactions of depths in the medium range.

An intuitive but somewhat naive alternative sampling process is to allow a variable to be included into an interaction according to its present "shallowness" quantified by $(i - 1 - M_{i-1})$ (instead of $M_{i-1}$ in the FFM). It can be verified that this construction will lead to a distribution of interactions which concentrates most its mass around $D/2$; moreover, exchangeability among $Z_i$ would be lost. To maintain exchangeability, we define the sampling process for the sequence $Z = (Z_1, \ldots, Z_D) \in \{0, 1\}^D$ as follows: let $\sigma(\cdot)$ be a random uniform permutation of $\{1, \ldots, D\}$ and let $\sigma_1 = \sigma^{-1}(1), \ldots, \sigma_D = \sigma^{-1}(D)$. Note that $\sigma_1, \ldots, \sigma_D$ are discrete random variables and $\mathbb{P}(\sigma_k = i) = 1/D$ for any $i, k = 1, \ldots, D$. For $i = 1, \ldots, D$, set

$$\mathbb{P}(Z_{\sigma_i} = 1 | Z_{\sigma_1}, \ldots, Z_{\sigma_{i-1}}) = \tfrac{\alpha M_{i-1} + (1-\alpha)(i-1-M_{i-1}) + \gamma_1}{i-1+\gamma_1+\gamma_2},$$

$$\mathbb{P}(Z_{\sigma_i} = 0 | Z_{\sigma_1}, \ldots, Z_{\sigma_{i-1}}) = \tfrac{(1-\alpha) M_{i-1} + \alpha(i-1-M_{i-1}) + \gamma_2}{i-1+\gamma_1+\gamma_2}, \quad (6)$$

where $\gamma_1 > 0, \gamma_2 > 0, \alpha \in [0, 1]$ are given parameters and $M_i = Z_{\sigma_1} + \ldots + Z_{\sigma_i}$. The collection of $Z$ generated by this process shall be called to follow $\text{FFM}_\alpha$. When $\alpha = 1$ we recover the original FFM prior. When $\alpha = 0$, we get the other extremal behavior mentioned at the beginning of the paragraph. Allowing $\alpha \in [0, 1]$ yields a richer spectrum spanning the two distinct extremal behaviors.

Details of the process and some of its properties are given in the Supplement. Here we briefly describe how $\text{FFM}_\alpha$ *a priori* ensures "poor gets richer" behavior and offers extra flexibility in modeling interaction depths compared to the original FFM. The depth of an interaction of $D$ variables is described by the distribution of $M_D$. Consider the conditionals obtained for a Gibbs sampler where index of a variable to be updated is random and based on $\mathbb{P}(\sigma_D = i | Z)$ (it is simply $1/D$ for $\text{FFM}_1$). Suppose we want to assess how likely it is to *add* a variable into an existing interaction via the expression $\sum_{i:Z_i^{(k)}=0} \mathbb{P}(Z_i^{(k+1)} = 1, \sigma_D = i | Z^{(k)})$, where $k + 1$ is the next iteration of the Gibbs sampler's conditional update. This probability is a function of $M_D^{(k)}$; for small values of $M_D^{(k)}$ it quantifies the tendency for the "poor gets richer" behavior. For the $\text{FFM}_1$ it is given by $\tfrac{D - M_D^{(k)}}{D} \tfrac{M_D^{(k)} + \gamma_1}{D - 1 + \gamma_1 + \gamma_2}$. In Fig. 1(a) we show that $\text{FFM}_1$'s behavior is opposite of "poor gets richer", while $\alpha \le 0.7$ appears to ensure the desired property. Next, in Fig.1 (b-f) we show the distribution of $M_D$ for various $\alpha$, which exhibits a broader spectrum of behavior.

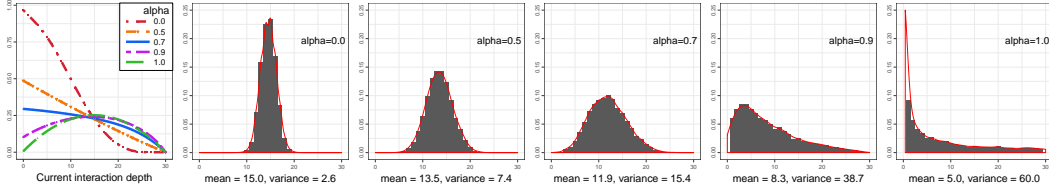

Figure 1: $D = 30$, $\gamma_1 = 0.2$, $\gamma_2 = 1$ (a) Probability of increasing interaction depth; (b-f) $\text{FFM}_\alpha$ $M_D$ distributions with different $\alpha$.

## 5 Experimental Results

### 5.1 Simulation Studies

We shall compare MiFM methods against a variety of other regression techniques in the literature, including Bayesian Factorization Machines (FM), lasso-type regression, Support Vector Regression (SVR), multilayer perceptron neural network (MLP).[2] The comparisons are done on the basis of prediction accuracy of responses (Root Mean Squared Error on the held out data), quality of regression coefficient estimates and the interactions recovered.

#### 5.1.1 Predictive Performance

In this set of experiments we demonstrate that MiFMs with either $\alpha = 0.7$ or $\alpha = 1$ have dominant predictive performance when high order interactions are in play.

In Fig. 2(a) we analyzed 70 random interactions of varying orders. We see that MiFM can handle arbitrary complexity of the interactions, while other methods are comparative only when interaction structure is simple (i.e. linear or 2-way on the right of the Fig. 2(a)).

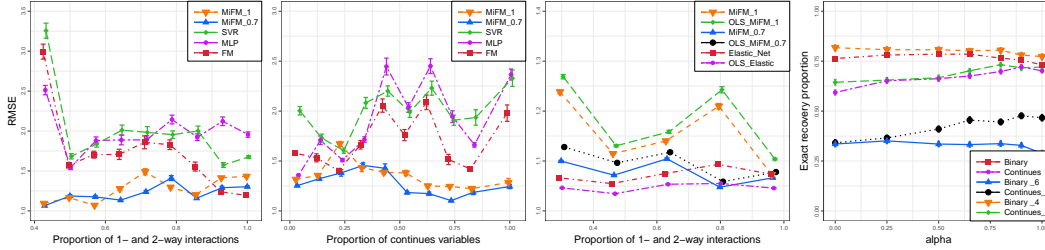

Figure 2: RMSE for experiments: (a) interactions depths; (b) data with different ratio of continuous to categorical variables; (c) quality of the $\text{MiFM}_1$ and $\text{MiFM}_{0.7}$ coefficients; (d) $\text{MiFM}_\alpha$ exact recovery of the interactions with different $\alpha$ and data scenarios

Next, to assess the effectiveness of MiFM in handling categorical variables (cf. Section 3.3) we vary the number of continuous variables from 1 (and 29 attributes across categories) to 30 (no categorical variables). Results in Fig. 2(b) demonstrate that our models can handle both variable types in the data (including continuous-categorical interactions), and still exhibit competitive RMSE performance.

### 5.1.2 Interactions Quality

**Coefficients of the interactions**    This experiment verifies the posterior consistency result of Theorem 1 and validates our factorization model for coefficients approximation. In Fig. 2(c) we compare MiFMs versus OLS fitted with the corresponding sets of chosen interactions. Additionally we benchmark against Elastic net (Zou & Hastie, 2005) based on the expanded data matrix with interactions of all depths included, that is $2^D - 1$ columns, and a corresponding OLS with only selected interactions.

**Selection of the interactions**    In this experiments we assess how well MiFM can recover true interactions. We consider three interaction structures: a realistic one with five linear, five 2-way, three 3-way and one of each $4, \ldots, 8$-way interactions, and two artificial ones with 15 either only 4- or only 6-way interactions to challenge our model. Both binary and continuous variables are explored. Fig. 2(d) shows that MiFM can *exactly* recover up to 83% of the interactions and with $\alpha = 0.8$ it recovers 75% of the interaction in 4 out of 6 scenarios. Situation with 6-way interactions is more challenging, where 36% for binary data is recovered and almost half for continuous. It is interesting to note that lower values of $\alpha$ handle binary data better, while higher values are more appropriate for continuous, which is especially noticeable on the "only 6-way" case. We think it might be related to the fact that high order interactions between binary variables are very rare in the data (i.e. product of 6 binary variables is equal to 0 most of the times) and we need a prior eager to explore ($\alpha = 0$) to find them.

## 5.2 Real world applications

### 5.2.1 Finding epistasis

Identifying epistasis (i.e. interactions between genes) is one of the major questions in the field of human genetics. Interactions between multiple genes and environmental factors can often tell a lot more about the presence of a certain disease than any of the genes individually (Templeton, 2000). Our analysis of the epistasis is based on the data from Himmelstein et al. (2011). These authors show that interactions between single nucleotide polymorphisms (SNPs) are often powerful predictors of various diseases, while individually SNPs might not contain important information at all. They developed a model free approach to simulate data mimicking relationships between complex gene interactions and the presence of a disease. We used datasets with five SNPs and either 3-,4- and 5-way interactions or only 5-way interactions. For this experiment we compared $\text{MiFM}_1$, $\text{MiFM}_0$; refitted logistic regression for each of our models based on the selected interactions ($\text{LMiFM}_1$ and $\text{LMiFM}_0$), Multilayer Perceptron with 3 layers and Random Forest.[3] Results in Table 1 demonstrate that MiFM produces competitive performance compared to the very best black-box techniques on this data set, while it also selects interacting genes (i.e. finds epistasis). We don't know which of the 3- and 4-way interactions are present in the data, but since there is only one possible 5-way interaction we can check if it was identified or not — both $\text{MiFM}_1$ and $\text{MiFM}_0$ had a 5-way interaction in at least 95% of the posterior samples.

Table 1: Prediction Accuracy on the Held-out Samples for the Gene Data

|  | MiFM$_1$ | MiFM$_0$ | LMiFM$_1$ | LMiFM$_0$ | MLP | RF |
|---|---|---|---|---|---|---|
| 3-, 4-, 5-way | 0.775 | 0.771 | 0.883 | 0.860 | 0.870 | 0.887 |
| only 5-way | 0.649 | 0.645 | 0.628 | 0.623 | 0.625 | 0.628 |

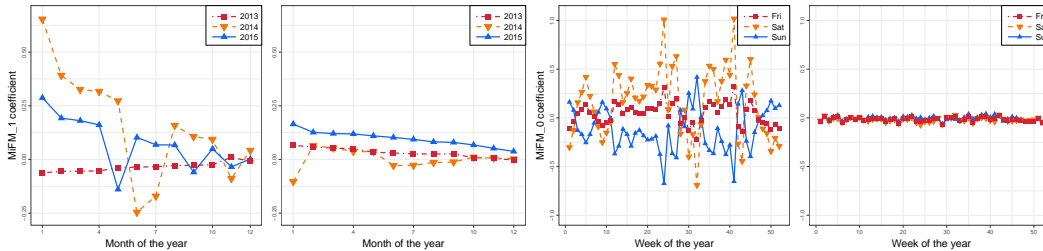

Figure 3: MiFM$_1$ store - month - year interaction: (a) store in Merignac; (b) store in Perols; MiFM$_0$ city - store - day of week - week of year interaction: (c) store in Merignac; (d) store in Perols.

### 5.2.2 Understanding retail demand

We finally report the analysis of data obtained from a major retailer with stores in multiple locations all over the world. This dataset has 430k observations and 26 variables spanning over 1100 binary variables after the one-hot encoding. Sales of a variety of products on different days and in different stores are provided as response. We will compare MiFM$_1$ and MiFM$_0$, both fitted with $K = 12$ and $J = 150$, versus Factorization Machines in terms of adjusted mean absolute percent error AMAPE $= 100 \frac{\sum_n |\hat{y}_n - y_n|}{\sum_n y_n}$, a common metric for evaluating sales forecasts. FM is currently a method of choice by the company for this data set, partly because the data is sparse and is similar in nature to the recommender systems. AMAPE for MiFM$_1$ is 92.4; for MiFM$_0$ - 92.45; for FM - 92.0.

**Posterior analysis of predictor interactions**  The unique strength of MiFM is the ability to provide valuable insights about the data through its posterior analysis. MiFM$_1$ recovered 62 non-linear interactions among which there are five 3-way and three 4-way. MiFM$_0$ selected 63 non-linear interactions including nine 3-way and four 4-way. We note that choice $\alpha = 0$ was made to explore deeper interactions and as we see MiFM$_0$ has more deeper interactions than MiFM$_1$. Coefficients for a 3-way interaction of MiFM$_1$ for two stores in France across years and months are shown in Fig. 3(a,b). We observe different behavior, which would not be captured by a low order interaction. In Fig. 3(c,d) we plot coefficients of a 4-way MiFM$_0$ interaction for the same two stores in France. It is interesting to note negative correlation between Saturday and Sunday coefficients for the store in Merignac, while the store in Perols is not affected by this interaction - this is an example of how MiFM can select interactions between attributes across categories.

## 6 Discussion

We have proposed a novel regression method which is capable of learning interactions of arbitrary orders among the regression predictors. Our model extends Finite Feature Model and utilizes the extension to specify a hypergraph of interactions, while adopting a factorization mechanism for representing the corresponding coefficients. We found that MiFM performs very well when there are some important interactions among a relatively high number (higher than two) of predictor variables. This is the situation where existing modeling techniques may be ill-equipped at describing and recovering. There are several future directions that we would like to pursue. A thorough understanding of the fully nonparametric version of the FFM$_\alpha$ is of interest, that is, when the number of columns is taken to infinity. Such understanding may lead to an extension of the IBP and new modeling approaches in various domains.

**Acknowledgments**

This research is supported in part by grants NSF CAREER DMS-1351362, NSF CNS-1409303, a research gift from Adobe Research and a Margaret and Herman Sokol Faculty Award.

## Footnotes

[1]Code is available at https://github.com/moonfolk/MiFM.

[2]Random Forest Regression and optimization based FM showed worse results than other methods.

[3]FM, SVM and logistic regression had low accuracy of around 50% and are not reported.

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
