[Supplementary Material]

# Supplementary Material for Multi-way Interacting Regression via Factorization Machines

**Mikhail Yurochkin**
Department of Statistics
University of Michigan
Ann Arbor, MI 48109
`moonfolk@umich.edu`

**XuanLong Nguyen**
Department of Statistics
University of Michigan
Ann Arbor, MI 48109
`xuanlong@umich.edu`

**Nikolaos Vasiloglou**
LogicBlox
Atlanta, GA 30309
`nikolaos.vasiloglou@logicblox.com`

In the Supplementary material we will start by proving consistency of the MiFM theorem, then we will show several important results related to $\text{FFM}_\alpha$: how exchangeability is achieved using uniform permutation prior on the order in which variables enter the process, how it leads to a Gibbs sampler using distribution of the index of the variable entering $\text{FFM}_\alpha$ last and how to obtain distribution of the interaction depths $M_D$ and compute its expectation. Lastly we will present a Gibbs sampling algorithm for the MiFM under the $\text{FFM}_\alpha$ prior on interactions structure $Z$.[1]

## 1 Proof of the Consistency Theorem

First let us remind the reader of the problem setup. Suppose that the data pairs $(x_n, y_n)_{n=1}^N \in \mathbb{R}^D \times \mathbb{R}$ are i.i.d. samples from the joint distribution $P^*(X, Y)$, according to which marginal distribution for $X$ and the conditional distribution of $Y$ given $X$ admit density functions $f^*(x)$ and $f^*(y|x)$, respectively, with respect to Lebesgue measure. In particular, $f^*(y|x)$ is defined by

$$Y = y_n | X = x_n, \Theta^* \sim \mathcal{N}(y(x_n, \Theta^*), \sigma), \text{ where } \Theta^* = \{\beta_1^*, \ldots, \beta_J^*, Z_1^*, \ldots, Z_J^*\},$$

$$y(x, \Theta^*) := \sum_{j=1}^J \beta_j^* \prod_{i \in Z_j^*} x_i, \text{ and } x_n \in \mathbb{R}^D, y_n \in \mathbb{R}, \beta_j^* \in \mathbb{R}, Z_j^* \subset \{1, \ldots, D\}, \tag{1}$$

$$\text{for } n = 1, \ldots, N, j = 1, \ldots, J.$$

In the above $\Theta^*$ represents the *true* parameter for the conditional density $f^*(y|x)$ that generates data sample $y_n$ given $x_n$, for $n = 1, \ldots, N$. On the other hand, the statistical modeler has access only to the MiFM:

$$Z \sim \text{FFM}_\alpha(\gamma_1, \gamma_2), \ v_{ik} | \mu_k, \lambda_k \sim \mathcal{N}(\mu_k, \frac{1}{\lambda_k}) \text{ for } i = 1, \ldots, D; \ k = 1, \ldots, K,$$

$$y_n | \Theta \sim \mathcal{N}(y(x_n, \Theta), \sigma), \text{ where } y(x, \Theta) := \sum_{j=1}^J \sum_{k=1}^K \prod_{i \in Z_j} x_i v_{ik}, \tag{2}$$

$$\text{for } n = 1, \ldots, N, \text{ and } \Theta = (Z, V).$$

We omitted linear terms in the MiFM since they can naturally be parts of the interaction structure $Z$ and discarded hyperpriors for the ease of representation. Now we show that under some conditions

posterior distribution $\Pi$ will place most of its mass on the true conditional density $f^*(y|x)$ as $N \to \infty$.

**Theorem 1.** Given any true conditional density $f^*(y|x)$ given by (1), and assuming that the support of $f^*(x)$ is bounded, there is a constant $K_0 < J$ such that by setting $K \geq K_0$, the following statement holds: for any weak neighborhood $U$ of $f^*(y|x)$, under the MiFM model (2), the posterior probability $\Pi(U|(X_n, Y_n)_{n=1}^N) \to 1$ with $P^*$-probability one, as $N \to \infty$.

A key part in the proof of this theorem is to clarify the role of parameter $K$, and the fact that under model (2), the regression coefficient $\beta_j$ associated with interaction $j$ is parameterized by $\beta_j := \sum_{k=1}^{K} \prod_{i \in Z_j} v_{ik}$, for $j = 1, \ldots, J$, which for some suitable choice of $\Theta = (Z, V)$ can represent exactly the true parameters $\beta_1^*, \ldots, \beta_J^*$, provided that $K$ is sufficiently large. The following basic lemma is informative.

**Lemma 1.** Let $m \in [1, J]$ be a natural number, $\beta_j \in \mathbb{R} \setminus \{0\}$ for $j = 1, \ldots, m$. Suppose that the $m$ subsets $Z_j \subset \{1, \ldots, D\}$ for $j = 1, \ldots m$ have non-empty intersection, then as long as $K \geq m$, the system of polynomial equations

$$\sum_{k=1}^{K} \prod_{i \in Z_j} v_{ik} = \beta_j, j = 1, \ldots, m \tag{3}$$

has at least one solution in terms of $v_{11}, \ldots, v_{DK}$ such that the following collection of $K$ vectors in $\mathbb{R}^m$, namely, $\{(\prod_{i \in Z_1} v_{ik}, \ldots, \prod_{i \in Z_m} v_{ik}), k = 1, \ldots, K\}$ contains $m$ linearly independent vectors.

*Proof.* Let $i_0$ be an element of the intersection of all $Z_j$, for $j = 1, \ldots, m$. We consider system (3) as linear with respect to $\{v_{i_0 1}, \ldots, v_{i_0 K}\}$, where corresponding coefficients are given by $\prod_{i \in Z_j \setminus \{i_0\}} v_{i,k}$, which we can pick to form a matrix of nonzero determinant. Hence by Rouché–Capelli theorem the system has at least one solution if $K \geq m$ and, since $\beta_j \neq 0$ for $\forall j$, the resulting $\{(\prod_{i \in Z_1} v_{ik}, \ldots, \prod_{i \in Z_m} v_{ik}), k = 1, \ldots, K\}$ contains at least $m$ linearly independent vectors. $\square$

**Lemma 2.** (This is Lemma 1 of the main text) Given natural number $J \geq 1$, $\beta_j \in \mathbb{R} \setminus \{0\}$ and $Z_j \subset \{1, \ldots, D\}$ for $j = 1, \ldots J$, exists $K_0 < J : \forall K \geq K_0$ system of polynomial equations (3) has at least one solution in terms of $v_{11}, \ldots, v_{DK}$.

*Proof.* The proof proceeds by performing an elimination process on the collection of variables $v_{ik}$ according to an ordering that we now define. Let $J_i = \text{card}(\{Z_j | i \in Z_j\})$ for $i = 1, \ldots, D$. Define $J^0 = \min_i J_i$ and $i_0 = \arg\min_i J_i$. If $K \geq J^0$ by Lemma 1 we can find a solution of the reduced system of equations

$$\sum_{k=1}^{K} \prod_{i \in Z_j} v_{i,k} = \beta_j, j \in \{j | i_0 \in Z_j\},$$

while maintaining the linear independence needed to apply Lemma 1 again further. Now we know that we can find a solution for equations indexed by $\{j | i_0 \in Z_j\}$. We remove them from system (3) and recompute $J^1 = \min_{i \neq i_0} J_i$ and $i_1 = \arg\min_{i \neq i_0} J_i$ to apply Lemma 1 again. Iteratively we will remove all the equations, meaning that there is at least one solution. Note that $J_i$ are decreasing since whenever we remove equations, number of $Z_j$s containing certain $i$ can only decrease. Therefore, we will need $K \geq K_0 := \max(J^0, J^1, \ldots, 0)$ in order to apply Lemma 1 on every elimination step. $\square$

From the proof of Lemma 2, it can be observed that $K_0 = \max(J^0, J^1, \ldots) \ll J$ when we anticipate only few interactions per variable, whereas the upper bound $K_0 = J - 1$ is attained when there are only $(D - 1)$-way interactions. Now we are ready to present a proof of the main theorem.

*Proof.* (of main theorem). By Lemma 2 and the fact that the probability of a finite number of independent continuous random vectors being linearly dependent is 0 it follows that under the MiFM

prior on $V$ as in (2) and $\forall \beta_1, \ldots, \beta_J \in \mathbb{R} \setminus \{0\}$, distinct $Z_1, \ldots, Z_J$ and $\epsilon > 0$

$$\Pi \left( \sum_{j=1}^{J} (\beta_j - \sum_k \prod_{i \in Z_j} v_{ik})^2 < \epsilon \,|\, Z_1, \ldots, Z_J \right) > 0. \tag{4}$$

From Eq. (8) it follows that for any $Z_1, \ldots, Z_J$, the prior probability of the corresponding incidence matrix is bounded away from 0. Combining this with (4), we now establish that the probability of the true model parameters to be arbitrary close to the MiFM parameters under the MiFM prior as in (2):

$$\Pi \left( (\sum_{j=1}^{J} \beta_j - \sum_{j=1}^{J} \sum_k \prod_{i \in Z_j} v_{ik})^2 < \epsilon \right) > 0, \ \forall \epsilon > 0. \tag{5}$$

We shall appeal to Schwartz's theorem (cf. Ghosal et al. (1999)), which asserts that the desired posterior consistency holds as soon as we can establish that the true joint distribution $P^*(X, Y)$ lies in the Kullback-Leibler support of the prior $\Pi$ on the joint distribution $P(X, Y)$. That is,

$$\Pi \left( \mathrm{KL}(P^* || P) < \epsilon \right) > 0, \ \text{for } \forall \epsilon > 0. \tag{6}$$

Since the KL divergence of the two Gaussian distributions is proportional to the mean difference, we have ($\mathbb{E}_X^*$ denotes expectation with respect to the true marginal distribution of $X$)

$$\mathrm{KL}(P^* || P) \propto \mathbb{E}_X^* \frac{1}{2} (y(X, \Theta) - y(X, \Theta^*))^2 \propto$$

$$\mathbb{E}_X^* (\sum_{j=1}^{J} \beta_j \prod_{i \in Z_j} x_i - \sum_{j=1}^{J} \sum_k \prod_{i \in Z_j} v_{ik} x_i)^2 \lesssim (\sum_{j=1}^{J} \beta_j - \sum_{j=1}^{J} \sum_k \prod_{i \in Z_j} v_{ik})^2. \tag{7}$$

Due to (5) this quantity can be made arbitrarily close to 0 with positive probability. Therefore (6) and then Schwartz theorem hold, which concludes the proof. $\qquad \square$

## 2 Analyzing FFM$_\alpha$

### 2.1 Model definition and exchangeability

Here we remind the reader the construction of FFM$_\alpha$ — the distribution over finite collection of binary random variables that we used to model interactions. Let $D$ be the number of variables in the data and $Z \in \{0, 1\}^D$ is $j$-th interaction (subscript $j$ is dropped to simplify notation). Let $\sigma(\cdot)$ be a random uniform permutation of $\{1, \ldots, D\}$ and let $\sigma_1 = \sigma^{-1}(1), \ldots, \sigma_D = \sigma^{-1}(D)$. Note that $\sigma_1, \ldots, \sigma_D$ are discrete random variables and $\mathbb{P}(\sigma_k = i) = 1/D$ for any $i, k = 1, \ldots, D$. Next we define FFM$_\alpha$:

$$\mathbb{P}(Z_{\sigma_i} = 1 | Z_{\sigma_1}, \ldots, Z_{\sigma_{i-1}}) = \frac{\alpha M_{i-1} + (1-\alpha)(i-1-M_{i-1}) + \gamma_1}{i-1+\gamma_1+\gamma_2},$$

$$\mathbb{P}(Z_{\sigma_i} = 0 | Z_{\sigma_1}, \ldots, Z_{\sigma_{i-1}}) = \frac{(1-\alpha) M_{i-1} + \alpha(i-1-M_{i-1}) + \gamma_2}{i-1+\gamma_1+\gamma_2}, \tag{8}$$

where $\gamma_1 > 0, \gamma_2 > 0, \alpha \in [0, 1]$ are given parameters and $M_i = Z_{\sigma_1} + \ldots + Z_{\sigma_i}$. Due to the random permutation of indices, distribution of $Z_1, \ldots, Z_D$ is exchangeable because any ordering of variables entering the process has same probability. Next, we need to integrate the permutation part out to obtain a tractable full conditional representation.

### 2.2 Gibbs sampling for FFM$_\alpha$ and distribution of interaction depths $M_D$

To construct a Gibbs sampler for the the FFM$_\alpha$ we will use an additional latent variable - index of the variable entering the process last, $\sigma_D$. Additionally observe that when permutation is integrated out $\mathbb{P}(Z_1, \ldots, Z_D) = \mathbb{P}(M_D = Z_1 + \ldots + Z_D)$ since $\mathbb{P}(M_D = m)$ is precisely the summation over all possible orderings of $Z_1, \ldots, Z_D$ such that $Z_1 + \ldots + Z_D = m$.

$$\mathbb{P}(\sigma_D = i | Z_1, \ldots, Z_D) \propto$$

$$Z_i \mathbb{P}(\sigma_D = i | Z_{\sigma_D} = 1, Z) \mathbb{P}(Z_{\sigma_D} = 1 | M_{D-1} = \sum_{k=1}^{D} Z_k - 1) \mathbb{P}(M_{D-1} = \sum_{k=1}^{D} Z_k - 1) +$$

$$+ (1 - Z_i) \mathbb{P}(\sigma_D = i | Z_{\sigma_D} = 0, Z) \mathbb{P}(Z_{\sigma_D} = 0 | M_{D-1} = \sum_{k=1}^{D} Z_k) \mathbb{P}(M_{D-1} = \sum_{k=1}^{D} Z_k), \tag{9}$$

then if $Z_i = 1$ and $\sum_{k=1}^{D} Z_k = m$ we obtain

$$\mathbb{P}(\sigma_D = i | Z_{-i}, Z_i = 1) = \mathbb{P}(\sigma_D = i | M_D = m, Z_i = 1) =$$
$$= \frac{\mathbb{P}(M_{D-1} = m - 1)\mathbb{P}(Z_{\sigma_D} = 1 | M_{D-1} = m - 1)}{m\mathbb{P}(M_D = m)}, \qquad (10)$$

where $\mathbb{P}(Z_{\sigma_D} = 1 | M_{D-1} = m - 1)$ and $\mathbb{P}(Z_{\sigma_D} = 0 | M_{D-1} = m)$ can be computed as in Eq. 8. Our next step is to analyze probability $\mathbb{P}(M_D = m)$. Indeed it is easy to obtain this distribution recursively:

$$\mathbb{P}(M_D = m) = \mathbb{P}(M_{D-1} = m)\mathbb{P}(Z_{\sigma_D} = 0 | M_{D-1} = m) +$$
$$+ \mathbb{P}(M_{D-1} = m - 1)\mathbb{P}(Z_{\sigma_D} = 1 | M_{D-1} = m - 1). \qquad (11)$$

The base of recursion is given by the following identities:

$$\mathbb{P}(M_0 = 0) = 1,$$
$$\mathbb{P}(M_i = 0) = \prod_{k=0}^{i-1} \frac{\alpha(i - 1 - k) + \gamma_2}{k + \gamma_1 + \gamma_2} = \prod_{k=0}^{i-1} \frac{\alpha k + \gamma_2}{k + \gamma_1 + \gamma_2}, \qquad (12)$$
$$\mathbb{P}(M_i = i) = \prod_{k=0}^{i-1} \frac{\alpha k + \gamma_1}{k + \gamma_1 + \gamma_2}.$$

The above formulation allows us compute $\mathbb{P}(M_i = k), D \geq i \geq k$ dynamically (computations are very fast since we only need to perform $\frac{(D+1)(D+2)}{2} - 1$ calculations) *before* running MiFM inference and utilize the table of probabilities during it. The last step of the Gibbs sampler is clearly the update of the $Z_i | \sigma_D = i, Z_{-i}$ which is done simply using the FFM$_\alpha$ definition 8. Recall Figure 1 (a) of the main text which illustrates the behavior of

$$\sum_{i:Z_i^{(k)}=0} \mathbb{P}(Z_i^{(k+1)} = 1, \sigma_D = i | Z^{(k)}) = \mathbb{P}(Z_{\sigma_D} = 0 | Z)\mathbb{P}(Z_i = 1 | \sigma_D = i, Z_{-i}),$$

and since we choose index of a variable to update based on the probability of it being last, the expression above reads as the probability that we choose to update a variable not present in the interaction and then add it to the interaction, therefore increasing the depth of the interaction.

## 2.3 Mean Behavior of the FFM$_\alpha$

From Eq. (11) it follows that

$$\mathbb{E}M_D = \sum_{m=0}^{D} m\mathbb{P}(M_D = m) =$$
$$= \frac{1}{D - 1 + \gamma_1 + \gamma_2}\Big\{ (1 - 2\alpha)\mathbb{E}M_{D-1}^2 + (\alpha(D - 1) + \gamma_2)\mathbb{E}M_{D-1} +$$
$$+ (2\alpha - 1)\mathbb{E}(M_{D-1} + 1)^2 + ((1 - \alpha)D - \alpha + \gamma_1)\mathbb{E}(M_{D-1} + 1) \Big\} \qquad (13)$$
$$= \frac{1}{D - 1 + \gamma_1 + \gamma_2}\Big\{ \mathbb{E}M_{D-1}(D + 2\alpha + \gamma_1 + \gamma_2 - 2) + D(1 - \alpha) + \alpha + \gamma_1 - 1 \Big\}.$$

For $\alpha = 0$, this relation is simplified to be

$$(D - 1 + \gamma_1 + \gamma_2)\mathbb{E}M_D = \mathbb{E}M_{D-1}(D + \gamma_1 + \gamma_2 - 2) + (D + \gamma_1 - 1) =$$
$$= (D + \gamma_1 - 1) + \ldots + \gamma_1 = \frac{1}{2}D(D + 2\gamma_1 - 1). \qquad (14)$$

# 3 Gibbs Sampler for the MiFM

Our Gibbs sampling algorithm consists of two parts — updating factorization coefficients $V$ (based on the results from Freudenthaler et al. (2011)) and then updating interactions $Z$ based on the analysis

of Section 2.2. Recall the MiFM model construction. First we have a layer of hyperpriors:

$$\sigma \sim \Gamma(\frac{\alpha_1}{2}, \frac{\beta_1}{2}), \qquad \lambda \sim \Gamma(\frac{\alpha_0}{2}, \frac{\beta_0}{2}), \qquad \mu \sim \mathcal{N}(\mu_0, \frac{1}{\gamma_0}),$$

$$\lambda_k \sim \Gamma(\frac{\alpha_0}{2}, \frac{\beta_0}{2}), \; \mu_k \sim \mathcal{N}(\mu_0, \frac{1}{\gamma_0}) \text{ for } k = 1, \dots, K,$$

Then interactions and their weights:

$$w_i | \mu, \lambda \sim \mathcal{N}(\mu, \frac{1}{\lambda}) \text{ for } i = 0, \dots, D, \qquad Z \sim \text{FFM}_\alpha(\gamma_1, \gamma_2),$$

$$v_{ik} | \mu_k, \lambda_k \sim \mathcal{N}(\mu_k, \frac{1}{\lambda_k}) \text{ for } i = 1, \dots, D; \; k = 1, \dots, K,$$

And finally the model's likelihood:

$$y_n | \Theta \sim \mathcal{N}(y(x_n, \Theta), \frac{1}{\sigma}), \text{ where}$$

$$y(x, \Theta) := w_0 + \sum_{i=1}^{D} w_i x_i + \sum_{j=1}^{J} \sum_{k=1}^{K} \prod_{i \in Z_j} x_i v_{ik}, \tag{15}$$

$$\text{for } n = 1, \dots, N, \text{ and } \Theta = \{Z, V, \sigma, w_{0,\dots,D}\}.$$

Inference in the context of Bayesian modeling is often related to learning the posterior distribution $\mathbb{P}(\Theta | X, Y)$. Then, if one wants point estimates, certain statistics of the posterior can be used, i.e. mean or median. In most situations (including MiFM) analytical form of the posterior is intractable, but with the help of Bayes rule it is often possible to compute it up to a proportionality constant:

$$\mathbb{P}(\Theta, \mu, \gamma, \mu_1, \dots \mu_K, \lambda_1, \dots, \lambda_K | Y) \propto \prod_{n=1}^{N} \mathbb{P}(y_n | Z, V, \sigma, w_{0,\dots,D}) \cdot$$
$$\cdot \mathbb{P}(Z) \mathbb{P}(V | \mu_1, \dots, \mu_K, \lambda_1, \dots, \lambda_K) \mathbb{P}(\sigma, \mu, \gamma, \mu_1, \dots \mu_K, \lambda_1, \dots, \lambda_K). \tag{16}$$

One can maximize this quantity to obtain MAP estimate, but this is very complicated due to the combinatorial complexity of interactions in $Z$ and, additionally, often leads to overfitting. We use Gibbs sampling procedure for learning the posterior of our model. Due to normal-normal conjugacy and a priori independence of $Z$ and other latent variables, we can derive closed form full conditional (i.e. variable given all the rest and the data) distributions for each of the latent variables in the model.

**Updating hyperprior parameters**

$$\sigma \sim \Gamma\left(\frac{\alpha_1 + N}{2}; \frac{\sum_{n=1}^{N}(y_n - y(x_n, \Theta))^2 + \beta_1}{2}\right), \tag{17}$$

$$\lambda \sim \Gamma\left(\frac{\alpha_0 + D + 1}{2}; \frac{\sum_{i=0}^{D}(w_i - \mu)^2 + \beta_0}{2}\right), \tag{18}$$

$$\mu \sim \mathcal{N}\left(\frac{\sum_{i=0}^{D} w_i + \gamma_0 \mu_0}{D + 1 + \gamma_0}; \frac{1}{\lambda(D + 1 + \gamma_0)}\right), \tag{19}$$

$$\lambda_k \sim \Gamma\left(\frac{\alpha_0 + D}{2}; \frac{\sum_{i=1}^{D}(v_{ik} - \mu_k)^2 + \beta_0}{2}\right), \tag{20}$$

$$\mu_k \sim \mathcal{N}\left(\frac{\sum_{i=1}^{D} v_{ik} + \gamma_0 \mu_0}{D + \gamma_0}; \frac{1}{\lambda_k(D + \gamma_0)}\right), \tag{21}$$
$$\text{for } k = 1, \dots, K.$$

**Updating factorization coefficients** $V$  For updating coefficients of the model we can utilize the multi-linear property also used for the Factorization Machines MCMC updates (Freudenthaler et al., 2011). Note that for any $\theta \in \{w_0, \dots, w_D, v_{11}, \dots, v_{DK}\}$ we can write $y(x, \Theta) = l_\theta(x) + \theta m_\theta(x)$, where $l_\theta(\cdot)$ are all the terms independent of $\theta$ and $m_\theta(\cdot)$ are the terms multiplied by $\theta$. For example, if $\theta = w_0$, then $m_\theta(x) = 1$ and $l_\theta(x) = \sum_{i=1}^{D} w_i x_i + \sum_{j=1}^{J} \sum_{k=1}^{K} \prod_{i \in Z_j} x_i v_{ik}$. Next we give

updating distribution that can be used for any $\theta \in \{w_0, \ldots, w_D, v_{11}, \ldots, v_{DK}\}$.

$$\theta \sim \mathcal{N}(\mu_\theta^*, \sigma_\theta^2), \text{where } \sigma_\theta^2 = \left( \sigma \sum_{n=1}^{N} m_\theta(x_n)^2 + \lambda_\theta \right)^{-1},$$

$$\mu_\theta^* = \sigma_\theta^2 \left( \sigma \sum_{n=1}^{N} (y_n - l_\theta(x_n)) m_\theta(x_n) + \mu_\theta \lambda_\theta \right), \tag{22}$$

and $\mu_\theta, \lambda_\theta$ are the corresponding hyperprior parameters.

**Updating interactions** $Z$    Posterior updates of $Z$ can be decomposed into prior times the likelihood:

$$\mathbb{P}(Z_i | Z_{-i}, V, Y) \propto \mathbb{P}(Z_i | Z_{-i}) \mathbb{P}(Y | V, Z), \tag{23}$$

where second part is the Gaussian likelihood as in Eq. (15). To sample $Z_i | Z_{-i}$ we use the construction from Section 2, where we first sample the value of $Z_{\sigma_D}$ for fixed $j$:

$$\mathbb{P}(Z_{\sigma_D} = 1 | Z) = \mathbb{P}(\sigma_D = i | M_D = m, Z_i = 1) =$$
$$= \frac{\mathbb{P}(M_{D-1} = m - 1) \mathbb{P}(Z_{\sigma_D} = 1 | M_{D-1} = m - 1)}{\mathbb{P}(M_D = m)}, \tag{24}$$

and then uniformly choose and index $i$ to update among $\{i : Z_i = Z_{\sigma_D}\}$. Next $Z_i$ can simply be updated using the process construction 8 assuming it to be last. Recall that $\mathbb{P}(M_D = m)$ should be computed beforehand using Eq. (11).

## Footnotes

[1]Code is available at `https://github.com/moonfolk/MiFM`.