[Reviews · NeurIPS 2017]

Reviewer 1



The authors present a regression method to learn interactions among arbitrary number of predictors. The prior on the multi-way interactions generalizes a parametric variant of the Indian Buffet process, which is then combined with Bayesian Factorization Machine (Rendle, 2010), to develop their MiFM model. They establish the posterior consistency of the model, develop and extend their priors for interactions and evaluate their predictive performance on both simulated and real data-sets. In simulations and real gene-interaction data sets, their method is shown to work very well in finding interactions among high number of predictors (e.g., in finding 5-way gene interactions). However, their predictive accuracy on held-out gene data still lags behind the top state-of-the-art methods in finding epistasis when all types of gene interactions (3-,4-,5- way) are considered. A particular strength of their method is shown to be in exploratory analysis, e.g., in selecting the genes with interactions. The authors describe improvements to the prior distributions on multi-way interactions, but this needs to be tied better to evaluation of the corresponding inferences in section 5. It's unclear to me if these improvements make a big difference in practice or not.

Reviewer 2



In this paper the authors propose a probabilistic model and an inference method for selecting high order interactions in a regression context. Unfortunately the paper is not directly in my field of expertise, therefore I cannot evaluate the novelty of the approach nor the fairness of the experiments; I am only able to provide high level comments. The paper overall is well-written. The motivation, the proposed model and the algorithm are stated very clearly. I like the model construction and the idea of adopting a probabilistic perspective for such a problem, which has not been explored in this field as the authors claim. The authors analyze the posterior consistency of the proposed model. Even though this theoretical result indicates that the approach is promising; however it is only asymptotic and not very informative for practical purposes. The experimental results are not very impressive, but still convincing for such a principled approach. The authors should provide information about the computational complexity of the approach and report the running time of their algorithm as well as the ones of the competitors.

Reviewer 3



The paper presents a Bayesian method for regression problems where the response variables can depend on multi-way combinations of the predictors. Via a hyper graph representation of the covariates interactions, the model is obtained from a refinement of the Finite Feature Model. The idea of modelling the interactions as the hyper edges of a hyper graph is interesting but the proposed model seems to be technically equivalent to the original Finite Mixture Model. Moreover, from the experimental results, it is hard to assess if the new parametrised extension is an improvement respect to the baseline. Here are few questions and comments: - is the idea of modelling interactions via an unbounded membership model new? Finite Mixture Models have been already used for predicting proteins interactions in [1], what are the key differences between that work and the proposed method? - the main motivation for modelling nonlinear interactions via multi-way regression is the interpretability of the identified model. Given the data, can one expect the nonlinear realisation to be unique? I think that a brief discussion about the identifiability of the model would improve the quality of the paper. - can the model handle higher powers of the same predictor (something like y = z x^2)? - the number of true interactions is usually expected to be very small respect to the number of all possible interactions. Does the model include a sparsity constraint on the coefficients/binary matrices? Is the low-rank assumption of the matrix V related to this? - is the factorised form of the multi-way interactions (matrix V in equation 3) expected to be flexible enough? - In the experiments: i) the true coefficients were chosen at random? Was there a threshold on their magnitude? ii) how is the `quality’ of the recovered coefficients defined? What is the difference between this and the `exact recovery’ of the interactions? iii) the standard FMM model (alpha=1) performs quite well independently of the number of multi-way interactions. Was that expected? iv) why does the random forest algorithm perform so well in the gene dataset and badly on the synthetic dataset? Why was the baseline (Elastic net algorithm) not used on the real-world experiments? ———————————————— [1] Krause et al. 2006 Identifying protein complexes in high-throughput protein interaction screens using an infinite latent feature model

Reviewer 4



This paper presents an extension to factorization machines (FM) to include Beta-Bernoulli priors for the presence of interaction terms of varying orders. The goal is to consider “methods that can adapt to fewer interactions but of arbitrarily varying orders” as opposed to limiting the maximum order of interactions to 2 or 3-way, as is typically done with FM. The model is presented and posterior consistency is discussed. The experimental section considers synthetic data with known interactions, SNP genetic analysis, and retail data, showing an effective and competitive method. It was an enjoyable read and the new method is innovative and certainly relevant to the recommendation systems research. I think it should be accepted because it presents a novel idea that is considered in depth, and is likely to make an impact to future research. A few additional comments on the paper: -there is still a hyperparameter dealing with the maximum order of interactions int he MiFM, to what extent is the method sensitive to this hyperparameter? -line 94 says that the model complexity grows exponentially with d=2 or 3, but the whole point of using an FM in the first place is that this grows linearly not exponentially; was line 94 referring to some other type of model? -the comment about the removal of the linear terms (line 128) was not so clear to me, surely the linear terms are needed? -the generative process of the model (Section 3.2) could have been presented more clearly; specifically, the layout and indexing makes it hard to see what the connection is between the weights, Z, and y(x, \Theta) -there were grammatical and spelling errors throughout, please proof read (e.g., “etc” —> “etc.”, “continues” —> “continuous”)